# ACTIVE TEST TIME PROMPT LEARNING IN VISION-LANGUAGE MODELS

## ABSTRACT

Test Time Optimisation is a setting where a model is made to learn new parameters on-the-fly during inference with the help of those very samples it is supposed to be tested on. Learning prompts at test time to improve the performance of Vision Language Models(VLMs) in downstream tasks has become a popular setting in recent times. In this paper, we propose a new framework for the Test Time Prompt Tuning in Pre-trained VLMs which incorporates actively sampled labels in the learning process to improve the performance of the model in downstream test-time settings. Our problem setting is underexplored yet well-motivated by considerations such as performance, efficiency and real-life applicability. Active Learning can be especially beneficial in the test-time setting in providing the option to query the true label when the model is uncertain in a real-life scenario and Prompt Tuning provides the advantage due to parameter efficiency. Our method is guided by these two principles and successfully combines the two to come up with a test-time optimisation scheme that is evaluated to be an improvement over existing methods under a fair evaluation protocol. We conduct experiments across 10 cross-dataset transfer datasets and 4 domain-generalisation datasets to show consistent improvement over the state-of-the-art.

## 1 INTRODUCTION

The development of large Vision Language Models (VLMs) (Radford et al., 2021; Li et al., 2022; 2019; Fini et al., 2023) has led to a paradigm shift in visual scene understanding that has traditionally been limited by a closed set of concepts seen during training. Through large-scale vision-language pre-training, these models learn to align language and vision modalities and show remarkable zero-shot transferability to unseen downstream tasks. A natural language description of the new class, known as *prompt*, (*e.g.*, 'a photo of a class') is fed to the text encoder of the VLM which is compared with the visual features generated by the vision encoder. However, finding the best hand-crafted prompt is non-trivial and calls for a lot of domain-specific heuristics. Prompt learning (Zhou et al., 2022b;a; Wang et al., 2023) has emerged as an alternative where a few soft prompts are learned for downstream tasks after freezing the entire VLM.

Traditionally, prompt learning has been supervised (Zhou et al., 2022a;b; Khattak et al., 2023) where prompts are trained on a labelled training dataset. Such approaches are naturally constrained by the availability of training data with annotations. Although pre-trained models are easily available nowadays, training data may not be available due to privacy, storage, or financial constraints. Moreover, in the case of a quick deployment scenario, it may not be possible to wait long to collect and annotate data for the downstream task as inference must continue. To address this challenge, Test-Time Transfer (TTT) (Liang et al., 2024; Wang et al., 2020; Yuan et al., 2023) has emerged as a promising approach. In the absence of the labelled data, the model is updated using an unsupervised objective. For example, (Shu et al., 2022) updates the prompts so that the average entropy of the logits from a set of augmentations of a single test example is minimized. However, the lack of labelled examples may not quite bridge the performance gap between the test time and fully supervised learning. In order to address this shortcoming, active learning (Ren et al., 2021; Zhan et al., 2022) algorithms can be used to incrementally select samples for annotation that improves performance with non-zero but low labelling cost.

We explore a new take on generalization at test time that takes a step back when the model is not quite certain about its decision and queries a human annotator for the true label of the uncertain test examples. Once queried, our framework optimizes the prompts using the examples with true labels in addition to the unlabelled test examples.

**Comparison with prior works.** A contemporary work ATTA (Gui et al., 2024) shows the impact of active learning in the standard TTA setting. Their analysis established more generalization ability in the model than standard TTA setting. However, their methodology and framework cannot be directly applied to large VLMs like CLIP (Radford et al., 2021) because they do not focus on the adaptation of a specific parameter group in a large model, while ours does so by incorporating the parameter-efficient Prompt Tuning. Further, they assumed a batch setting whereas we have a single test sample at a time and they also did multiple gradient updates per minibatch while we have only one gradient update per sample. As our approach updates and improves the model after encountering a single test example without waiting for additional examples to arrive, it is more flexible in continuous data streaming scenarios (Zhang et al., 2022). There is another recent work (Bang et al., 2024) which has used Prompt Learning and Active Learning together, but that was not in a test-time setting (which makes our problem more challenging), and they also did not try on adaptation tasks. For more details on related work check section B of supplementary material.

**Motivation.** All of the above facts strongly motivate our problem-setting. Foundation models are natural candidates to be used in a test time setting, and there are multiple use cases in daily life where a model could be so uncertain about something that instead of making a wrong prediction, it is better to query an expert whom we can abstract as an oracle (Zhu & Nowak, 2022; Shekhar et al., 2021). Examples include autopilot systems and medical applications. In such use cases, the extra cost and latency incurred in consulting the oracle is often worth it owing to greater generalisability and accuracy (supplementary section C). We have further motivated our approach by demonstrating a shortcoming of the existing TPT methodology through an experiment in section 3.

**Challenges and Novelty.** Prior work (Gui et al., 2024) had a batch setting and thus could simply actively sample a fixed proportion of the batch. However, in our case we only have a single sample at a time and have to make the decision of whether to sample it or not without knowing how uncertain the future samples would be. So we choose to dynamically adjust our threshold which would determine whether or not to actively label a sample. We also store the actively labelled samples in a fixed size buffer which would enable us to fully extract information from the samples, which would not be possible in a single time step. When the buffer is full, non-informative samples from over-represented classes are evicted from them. In adaptation tasks we follow (Abdul Samadh et al., 2024) to introduce distribution alignment, except that we align active samples to source data statistics of the particular class they belong to instead of the overall source data. As this is a novel problem setting, we have to make the effort to define an evaluation protocol which is sensible and fair. We do so, and it is shown that the evaluation protocol allows us to compare with previous arts without any unfairness.

In summary, our contributions are the following-

1. We propose a novel approach in the test-time optimisation of Vision Language models where we are equipped with the ability to query for the true labels to mitigate risk and improve generalisation capacity of the model. To the best of our knowledge, this is a novel setting and we are the first to apply Active Learning in a Test Time setting for VLMs.

2. We use the innovative idea of using a dynamically adjusted threshold to decide which test time samples have to be queried. We also incorporate the idea of class balancing in the annotation buffer and the replacement of non-informative samples which ablative studies reveal to be crucial to our performance.

3. For adaptation tasks, we also use the novel idea of class aware distribution alignment which makes effective use of the actively labelled samples to achieve more fine-grained distribution alignment.

4. The proposed algorithm performs better than many other methods under a low annotation budget and a limited buffer capacity. The evaluation protocol under which it was evaluated is demonstrably fair.

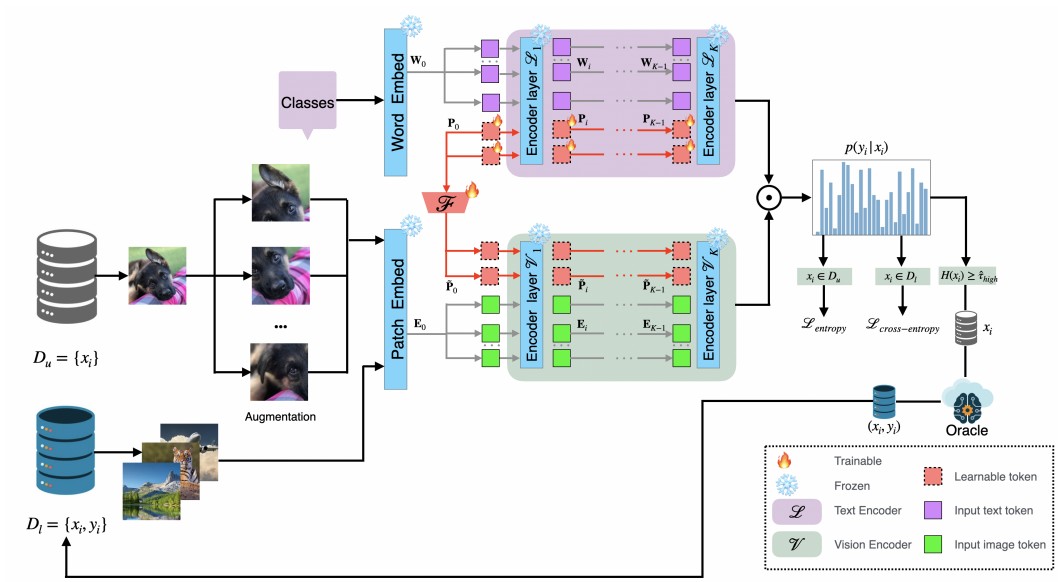

Figure 1: Overview of our method. We query uncertain samples at test time and put them in a buffer of limited size. We decide to query a sample if its entropy is above a certain threshold ($\tau_{\text{high}}$) which is dynamically adjusted. The old samples of a disproportionately represented class in the buffer are removed when they are not informative anymore.

## 2 PRELIMINARIES

### 2.1 ACTIVE LEARNING

In active learning, we have an unlabelled dataset $\mathcal{D}_u$. The typical setting is that of multi-class classification, having $K$ classes. The training happens in an iterative way where, in each iteration, the model selects some samples from $\mathcal{D}_u$, which it then passes on to the oracle to get the correct annotations. At each iteration, labelled samples are added to the labelled dataset $\mathcal{D}_l$. The labelled dataset is used to train the model before the next query phase, after which the size of $\mathcal{D}_l$ further increases. There is typically also a budget constraint that the model has to maintain. That is, the total number of queried samples cannot exceed a budget of $B$.

### 2.2 VISION LANGUAGE MODELS AND PROMPT LEARNING

In Vision Language Models (VLMs) like CLIP (Radford et al., 2021), training is performed by pairing an image and the corresponding caption. There are two encoders - an image encoder $E_v$ and a text encoder $E_t$. The image and text are passed through the encoders where the image encoder is either ViT (Dosovitskiy et al., 2020) or ResNet (He et al., 2016) architectures while the text encoder uses the Transformer (Vaswani et al., 2017; Devlin et al., 2018) architecture.

Once the image $x$ and text input $t_i$ (which is typically a classname out of the $K$ classnames) are mapped into the embedding space, which can be represented as

$$e_v = E_v(x) \qquad e_t^i = E_t(\{t_i; p\})$$

where $p$ is a handcrafted prompt prepended to the text input. Next, cosine similarity is computed between the visual embedding $e_v$ and textual embedding $e_t$. The objective of contrastive training is to reduce the distance between correct pairings and increase that between incorrect ones. That also achieves the objective of ensuring that the following prediction probability distribution

$$P(c = i | x) = \frac{exp(cos(e_v, e_t^i)/\tau)}{\sum_{j=1}^{K} exp(cos(e_v, e_t^j)/\tau)}$$

concentrates more of the density towards the correct class. Here $cos$ denotes the cosine similarity. Prompt Learning essentially makes the optimisation of a pre-trained VLM easier by only optimis-

ing a small parameter group known as prompts which are auxiliary to the original VLM. Instead of handcrafted prompts, which may require a lot of manual tuning, prompts are used as learnable vectors. The prompts are concatenated to the text input which is then passed through the text encoder to be mapped into the embedding space. There also exists a variant of prompting known as Multimodal Prompt Learning (Khattak et al., 2023) where learnable vectors are attached to both visual and textual tokens. In this paper, we use multimodal prompt learning.

## 2.3 TEST TIME PROMPT TUNING

In TPT (Shu et al., 2022), they have the same prompt tuning setting except that the prompt tuning has to be done on the fly on a single test sample $x_t$ at a time $t$. They then use a set of augmentation functions $\mathcal{A}$ to make $n$ augmentations $\tilde{x}_t = \mathcal{A}(x_t)$. $\tilde{x}_t$ is then passed through $E_v$ to get the logits corresponding probability distribution over classes for each augmentation. The entropy for each of the $n$ logits is found, and only the top $\rho$ logits with the least entropy are retained. The logits are averaged and the entropy of the average logit is computed. The marginal entropy of this average logit is the training objective at each time step $t$. This is called Marginal Entropy Minimisation (MEM) as we try to minimise this objective.

$$\mathcal{L}_{\text{entropy}} = -\sum_{i=1}^{C} \tilde{p}_{\boldsymbol{p}}(y_i|X_{\text{test}}) \log \tilde{p}_{\boldsymbol{p}}(y_i|X_{\text{test}}), \tag{1}$$

where $\boldsymbol{p}$ are the learnable prompts and $\tilde{p}_{\boldsymbol{p}}(y_i|X_{\text{test}})$ represents the mean of vector class probabilities produced by the model across the different augmented views preserved after the confidence selection filter.

## 2.4 DISTRIBUTION ALIGNMENT USING MULTIMODAL PROMPTING

In (Abdul Samadh et al., 2024), they use ImageNet (Deng et al., 2009) as the proxy source dataset. They generate $N$ random views of the test samples using a set of augmentations $\mathcal{H}$. They compute the mean and variance statistics of the token embeddings of the test samples at the output of each transformer layer of the CLIP model's visual encoder across the $N$ views. Similarly, the source data statistics from the proxy source dataset were pre-computed in an offline manner. The test sample distribution is represented by $(\mathcal{T})$ and the source distribution by $(\mathcal{D})$. Specifically, the token means and variances for the alignment are computed as follows.

$$\boldsymbol{\mu}_l(\mathcal{T}; \boldsymbol{p}) = \frac{1}{N} \sum_{\text{x} \in \mathcal{H}(X)} \tilde{\boldsymbol{X}}_{l,\text{x}}^{\boldsymbol{p}} \quad, \tag{2}$$

$$\boldsymbol{\sigma}_l^2(\mathcal{T}; \boldsymbol{p}) = \frac{1}{N} \sum_{\text{x} \in \mathcal{H}(X)} \left( \tilde{\boldsymbol{X}}_{l,\text{x}}^{\boldsymbol{p}} - \boldsymbol{\mu}_l(\mathcal{T}; \boldsymbol{p}) \right)^2, \tag{3}$$

where $\boldsymbol{\mu}_l(\mathcal{T}; \boldsymbol{p})$ and $\boldsymbol{\sigma}_l^2(\mathcal{T}; \boldsymbol{p})$ are the vector means and variances of the test sample tokens at the layer $l$ in the visual encoder and $\tilde{\boldsymbol{X}}_{l,\text{x}}^{\boldsymbol{p}}$ represents the prompted token embeddings at layer $l$ for the augmented view input x. Similarly, for each layer $l$ in the visual encoder, the source data statistics are pre-computed as,

$$\hat{\boldsymbol{\mu}}_l = \boldsymbol{\mu}_l(\mathcal{D}, \theta_v) \quad \text{and} \quad \hat{\boldsymbol{\sigma}}_l^2 = \boldsymbol{\sigma}_l^2(\mathcal{D}, \theta_v) \quad, \tag{4}$$

where $\theta_v$ denotes the parameters of the visual encoder from the pre-trained CLIP model. The token distribution alignment loss between the mean and variances of the test sample and the source dataset statistics is computed using the following,

$$\mathcal{L}_{\text{align}} = \frac{1}{L} \sum_{l=1}^{L} \left( \|\boldsymbol{\mu}_l(\mathcal{T}; \boldsymbol{p}) - \hat{\boldsymbol{\mu}}_l\|_1 + \|\boldsymbol{\sigma}_l^2(\mathcal{T}; \boldsymbol{p}) - \hat{\boldsymbol{\sigma}}_l^2\|_1 \right). \tag{5}$$

As shown above, $L_1$ loss is used to enforce the distribution alignment of the test sample with the source distribution.

## 3 METHOD

One shortcoming with the approach in TPT is that in trying to increase the certainty and consistency of a test sample across views, it potentially learns spurious representations. This is due to the presence of samples for which it is generally uncertain, and imposing consistency and certainty on such samples will potentially make the model predict wrong labels with certainty. We confirm this hypothesis by doing an experiment where we do not update on those samples with high entropy and instead just evaluate on them and let them pass. The results are given in Table 1. To mitigate the learning of spurious representation via updation on highly uncertain samples and also to not waste the information that could be provided by them by removing them from the training scheme, we propose to actively query some samples for which the model is generally uncertain. We formalise this notion of general uncertainty by selecting those samples for which the entropy of the average logit is more than a certain threshold which is *dynamically* adjusted. Our approach is visualised in Figure 1. We tried a fair evaluation protocol, which we have described in section 4.2. The most crucial detail in an evaluation protocol is the point in time at which we solicit the true label from the oracle. In particular, when we solicit the label before evaluating the sample, we cannot have a fair evaluation. Thus, our evaluation makes sure we are not using the ground truth label from the oracle and still keeping the sample for evaluation. In samples that are not queried, we apply the standard unsupervised loss as marginal entropy of the average logit.

Table 1: Motivating findings of not updating on low confidence samples

| Dataset | CLIP+TPT | CLIP+TPT, not updating on top 2% |
|---|---|---|
| DTD | 47.75 | 47.80 |
| Stanford Cars | 66.87 | 66.93 |
| Oxford Pets | 87.79 | 87.85 |
| Food101 | 84.67 | 84.70 |

### 3.1 FORMALISATION

We have a buffer $\mathcal{D}_l$, which is initially empty and is used to store queried samples. The buffer size is assumed to be limited to ensure that the framework is realistic. Let $\mathcal{D}_u$ denote the entire dataset. At time $t$, a test sample $x_t \in \mathcal{D}_u$ arrives to be evaluated. The test sample is augmented $N$ times to produce a batch of augmentations $\tilde{x}_t$. Like TPT, we find the logits for each of them and calculate the marginal entropy. The noisy augmentations are thus discarded. The average of the remaining logits is computed and the entropy corresponding to that is found. We denote that entropy as $H(x_t)$. If $H(x_t) > \tau_h$, then we choose to query it.

**Dynamic Threshold Selection.** $\tau_h$ is dynamically adjusted. Let us denote the threshold at time step t as $\tau_h^t$.

$$\tau_h^t = \hat{\mu}_t + z\hat{\sigma}_t \tag{6}$$

Where $\hat{\mu}_t$ and $\hat{\sigma}_t$ denote the estimated mean and standard deviation of $H(x_t)$ upto time step $t$. For the first $\tilde{t}$ time steps, we keep the threshold static. That is, $\tau_h^t = \tau \; \forall t \in [\tilde{t}]$. Since we also want to be within our budget and not exhaust it too early in the test stream, we adjust the value of $z$ depending on the proportion of samples queried in the test stream up to that point. Our approach is summarised in Algorithm 1 in section D of the supplementary material.

**Class Balanced Replacement Policy.** As our buffer size is limited, we came up with a replacement policy which would help us choose which sample to remove from the buffer in favour of an incoming sample. In evicting samples from the buffer, we adopt a notion of diversity achieved via class balancing. That is, we ensure that all classes are well represented in $\mathcal{D}_l$. We remove the sample with the lowest cross-entropy loss in the class with the most number of samples. The rationale is - the class with the most number of samples is over-represented, and the sample with the least cross-entropy loss is not that informative anymore. Our approach is summarised in Algorithm 2 in section D of the supplementary material.

Our loss function is of the form

$$\mathcal{L} = \mathcal{L}_{entropy} + \alpha \mathcal{L}_{cross\text{-}entropy} + \beta \mathcal{L}_{align} + \gamma \mathcal{L}_{class\text{-}align} \tag{7}$$

Where $\mathcal{L}_{entropy}$ is expressed in equation 1 and $\mathcal{L}_{align}$ is expressed in equation 5 while $\mathcal{L}_{cross\text{-}entropy}$ is the standard supervised cross-entropy loss applied on actively labelled samples. It is also important to note that except for domain generalisation tasks, we make $\beta = 0$ and $\gamma = 0$; that is, we do not use explicit domain alignment. In the domain generalisation task, we do a finer-grained version of domain alignment for actively labelled samples wherein we don't just align them to the statistics of the entire source dataset but the class statistics of the particular class to which they belong. We denote the corresponding loss by $\mathcal{L}_{class\text{-}align}$. On other samples, we do vanilla distribution alignment with general source data statistics. The $\mathcal{L}_{class\text{-}align}$ expression would be similar to the one in equation 5 except that in the place of $\hat{\mu}_l$ and $\hat{\sigma}_l^2$ we will have $\hat{\mu}_{l,c}$ and $\hat{\sigma}_{l,c}^2$ which would denote the mean and variance of representations of datapoints in the class $c$ in the proxy source dataset.

## 4 EXPERIMENT

Table 2: Results on cross-dataset transfer - the Top-1 accuracy is reported

| | Caltech101 | OxfordPets | StanfordCars | Flowers | Food101 | FGVC-Aircraft | SUN397 | DTD | EUROSAT | UCF101 | Average |
|---|---|---|---|---|---|---|---|---|---|---|---|
| CLIP | 93.35 | 88.25 | 65.48 | 67.44 | 83.65 | 23.67 | 62.59 | 44.27 | 42.01 | 65.13 | 63.58 |
| CLIP+TPT | 94.16 | 87.79 | 66.87 | 68.98 | 84.67 | 24.78 | 65.50 | 47.75 | 42.44 | 68.04 | 65.10 |
| CoOp | 93.70 | 89.14 | 64.51 | 68.71 | 85.30 | 18.47 | 64.15 | 41.92 | 46.39 | 66.55 | 63.88 |
| CoCoOp | 93.79 | 90.46 | 64.90 | 70.85 | 83.97 | 22.29 | 66.89 | 45.45 | 39.23 | 68.44 | 64.63 |
| ProDA | 86.70 | 88.20 | 60.10 | 71.50 | 80.80 | 22.20 | - | 50.90 | 58.50 | - | 65.62 |
| MaPLe | 93.53 | 90.49 | 65.57 | 72.23 | 86.20 | 24.74 | 67.01 | 46.49 | 48.06 | 68.69 | 66.30 |
| MaPLe+TPT | 93.59 | 90.72 | 66.50 | 72.37 | 86.64 | 24.70 | 67.54 | 45.87 | 47.80 | 69.19 | 66.50 |
| PromptAlign | 94.01 | **90.76** | **68.50** | **72.39** | 86.65 | 24.80 | **67.54** | 47.24 | 47.86 | 69.47 | 66.92 |
| ours | **94.27** | 90.65 | 67.86 | 72.36 | **86.77** | **24.85** | 67.51 | **48.23** | **49.88** | **70.23** | **67.26** |

Table 3: Results on adaptation datasets - the zero-shot Top-1 accuracy is reported

| | Imagenet-V2 | ImageNet-Sketch | ImageNet-A | Imagenet-R | OOD Avg. |
|---|---|---|---|---|---|
| CLIP | 60.86 | 46.09 | 47.87 | 73.98 | 57.20 |
| CLIP+TPT | 64.35 | 47.94 | 54.77 | 77.06 | 60.81 |
| CoOP | 64.20 | 47.99 | 49.71 | 75.21 | 59.28 |
| CoOp+TPT | **66.83** | 49.29 | 57.95 | 77.27 | 62.84 |
| Co-CoOp | 64.07 | 48.75 | 50.63 | 76.18 | 59.91 |
| Co-Coop+TPT | 64.85 | 48.27 | 58.47 | 78.65 | 62.61 |
| PromptAlign | 65.29 | **50.23** | 59.15 | 79.02 | 63.42 |
| ours | 65.60 | 50.14 | **59.31** | **79.51** | **63.64** |

### 4.1 EXPERIMENTAL SETUP

**Datasets.** In domain generalization, we evaluate on four out-of-distribution (OOD) variants of ImageNet (Deng et al., 2009); ImageNet-Sketch (Wang et al., 2019),ImageNet-A (Hendrycks et al., 2021b), ImageNet-V2 (Recht et al., 2019) and ImageNet-R (Hendrycks et al., 2021a). For cross-dataset transfer, we try on 10 image classification datasets which cover a wide variety of visual recognition tasks. Among these Caltech101 (Fei-Fei et al., 2004); five datasets which are fine-grained StanfordCars (Krause et al., 2013),Flowers102 (Nilsback & Zisserman, 2008),Oxford-Pets (Parkhi et al., 2012),Food101 (Bossard et al., 2014) and FGVC-Aircraft (Maji et al., 2013), which contain images of transportation, flowers and animals; and four datasets of textures, satellite imagery, scenes and human actions which are DTD (Cimpoi et al., 2014), EUROSAT (Helber et al.,

2019), SUN397 (Sun et al., 2020) and UCF101 (Soomro et al., 2012) respectively.

**Implementation Details.** Following PromptAlign (Abdul Samadh et al., 2024), using a single test sample we optimize the prompts on both the text and vision branches. Our models were implemented on a single NVIDIA A40 48GB GPU using the PyTorch framework. Refer to section 2.3, we take $n = 63$ and $\rho = 10\%$. Then we compute the token distribution alignment loss between the tokens of all the 64 images(equation 5). A learning rate of $5e^{-4}$ was used for the fine-grained datasets Flowers102, OxfordPets, Food101, SUN397, FGVCAircraft, and EuroSAT and a learning rate of 0.004 for the rest of the datasets. We use an annotation budget of $5\%$ and a buffer size of 150 for all datasets except Imagenet-v2 and Imagenet-Sketch where we use a buffer size of 75 due to memory constraints. The static threshold $\tau$ for selecting samples to be queried for labeling was fixed at 2, till $\tilde{t} = 30$ time steps. All the results were obtained by taking the average performance over 3 seeds for each dataset. Refer to equation 7. All the results in Table 2 are produced by taking $\alpha = 1$, $\beta = 0$ and $\gamma = 0$. On the other hand, in Table 3, the values of $\alpha$ are 1 in Imagenet-R and Imagenet-V2, and 0.15 and 0.5 in Imagenet-A and Imagenet-Sketch respectively, $\beta = 1$ and $\gamma = \alpha$ for all. We hypothesize that the lower value of $\alpha$ suited for these datasets is due to a greater number of outlier samples present in them.

**Baselines.** We evaluate our method with existing few-shot prompt learning methods for adapting CLIP including CoOp (Zhou et al., 2022b) and CoCoOp (Zhou et al., 2022a), TPT (Shu et al., 2022) and PromptAlign (Abdul Samadh et al., 2024) method. MaPLe (Khattak et al., 2023) is a multi-modal prompt learning baseline, which adapts CLIP by learning deep prompts on both the text and vision branches. TPT is a test-time prompt tuning method that tunes the prompt at test time per input sample, which achieved strong performance in prompt learning when combined with CoOp. It is important to note that whenever we have appended +TPT to a method, it means that the corresponding supervised counterpart has been taken and executed with TPT loss.

## 4.2 EVALUATION PROTOCOL

In the standard TPT setting, with the MEM (Section 2.3) framework, there is an update on the unsupervised loss before the evaluation. In our setting, we also update on the unsupervised loss before evaluation, but we can't use a sample to calculate the supervised loss before evaluation since it doesn't make sense to evaluate a sample after already knowing its true label. In our evaluation protocol, we update on the unsupervised loss before evaluating on the sample, but actively label it only after the evaluation. After active labelling, the sample is put in the buffer and can be used in calculating the $\mathcal{L}_{cross\text{-}entropy}$ in equation 7 for the successive time steps. This ensures fairness in evaluation since we are using the true label to update the model only after a sample has been evaluated. This also means that we can evaluate prior arts with their usual evaluation scheme.

## 4.3 RESULTS

The results of fine-grained classification are given in Table 2 while those of domain generalisation are given in Table 3. It can be seen that in nearly all of the datasets in fine-grained classification we are exceeding the state-of-the-art PromptAlign(Abdul Samadh et al., 2024) in terms of performance. Our average performance exceeds that of PromptAlign by 0.34%. In domain generalisation, our average performance exceeds that of the SOTA PromptAlign by $0.22\%$. While in ImageNet R and ImageNet A we show better performance than the SOTA, in ImageNet Sketch our performance is marginally worse while in ImageNet-v2 our performance is better than PromptAlign but worse than the method CoOp+TPT. In (Abdul Samadh et al., 2024) they explain this disparity by hypothesizing this is due to the extensive training of CoOp on ImageNet which has a very similar distribution to ImageNet-V2.

## 4.4 ABLATION STUDIES

### 4.4.1 CLASS BALANCED VS UNBALANCED (RANDOM DELETION FROM $\mathcal{D}_l$)

For class balancing we used Algorithm 2, for a description of the class balancing method refer to section 3.1.

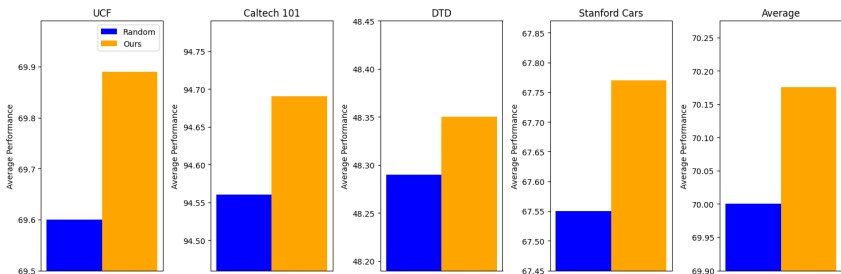

Figure 2: Class balanced vs random deletion from $\mathcal{D}_l$, our class balanced deletion policy performed better in all of the datasets

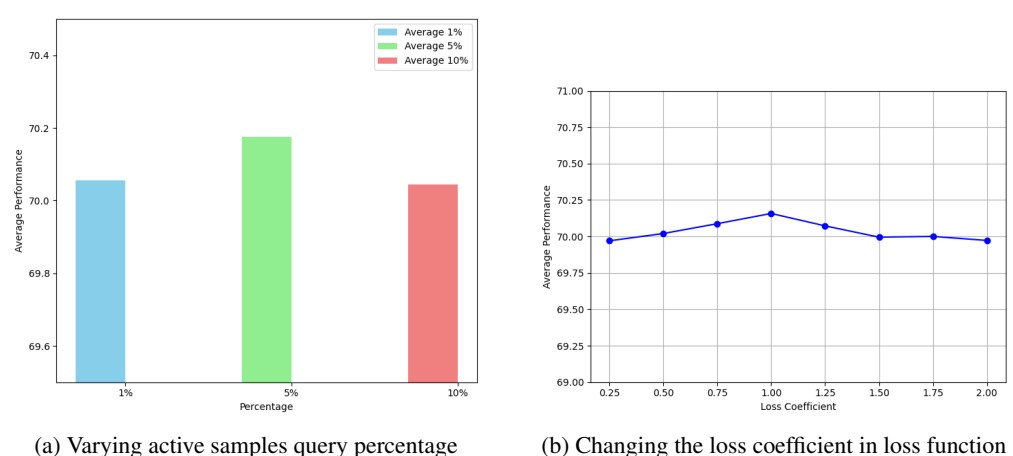

(a) Varying active samples query percentage    (b) Changing the loss coefficient in loss function

Figure 3: (a) 5% active query percentage was found to be optimum (b) The optimum value was found to be 1. The performance in both cases seemed to be quite robust to the change of the query percentage and change of loss coefficient respectively

For random deletion from $\mathcal{D}_l$, we randomly selected a class $c_k$ among all the classes such that $|c_k| \geq 1$, i.e. a non empty class, and then randomly deleted a sample from it.
It is evaluated on Caltech101, DTD, Stanford Cars, UCF101 and the results shown in Figure 2

### 4.4.2    ACTIVE SAMPLES QUERIED PERCENTAGE 1%,5%,10%:

Increasing the number of samples actively queried increases the model's robustness and hence helps improve its performance, especially for more challenging datasets. But that comes at a cost of the annotation budget, since the annotation budget is quite limited. We tried our experiments for different annotation budgets of 1%,5%,10%. The results of using different annotation budgets are given in Figure 3a which were obtained by taking the average performance over Caltech101, DTD, Stanford Cars and UCF101 datasets. The results show that increasing the annotation budget does not necessarily increase gains - there is an optimum annotation budget, which we found to be around 5%. We hypothesise that this is because of our limited buffer capacity which implies that with greater annotation budget, samples have to be replaced more often before all the information has been extracted from them. In any case, the performance is robust to change in query percentage as well - perhaps pointing to the fact that most of the information is contained in the top 1% (by entropy) of samples.

### 4.4.3    CHANGING THE LOSS COEFFICIENT IN LOSS FUNCTION

When we increase the value of $\alpha$ in equation 7, it means giving more importance to the unsupervised loss so we change the value of $\alpha$ in this range, the results are given in Figure 3b which were obtained by taking the average performance over Caltech101, DTD, Stanford Cars and UCF101 datasets. The

results show that on average, the value $\alpha = 1$ is the optimum value. However, the performance didn't seem to be very sensitive the value of the coefficient.

### 4.4.4 CHANGING THE SIZE OF $\mathcal{D}_l$(BUFFER) FOR VALUES 25,50,100,150:

Now the size of the buffer was varied and the corresponding performance was checked. We can not arbitrarily have a very large buffer size i.e. storing all the actively queried samples since for datasets like Food101, 5% of the total test samples is around 1500 images. Storing all those images in the buffer would increase the latency. So the maximum buffer size was set to 150. The results in Figure 4a were obtained by changing the buffer size to the values mentioned above, while 5% of the test samples were queried and taking the average performance for the datasets Caltech101, DTD, Stanford Cars, UCF101.

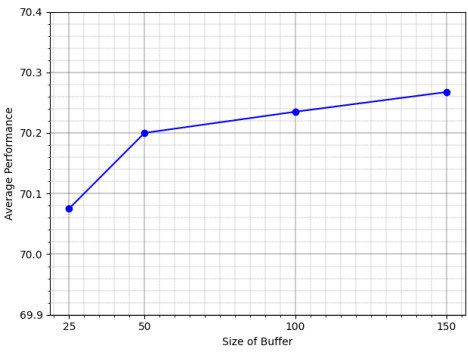
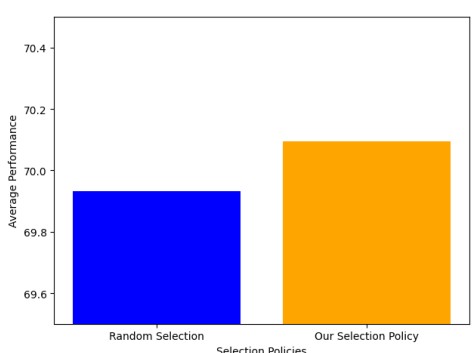

|  (a) Varying the size of the buffer  |  (b) Selecting random images vs our selection policy  |

Figure 4: (a) Increasing the buffer size leads to increase in average performance (b) Our image selection policy outperforms random image selection

### 4.4.5 SELECTING RANDOM IMAGES FOR THE BUFFER VS OUR SELECTION POLICY

Our selection policy of selecting informative samples from the test images for the buffer was now compared to selecting random images from the buffer. This was performed with maximum buffer size of 150, query of 5% test samples, and is the average of the performance in the Caltech101, DTD, Stanford Cars and UCF101 datasets. The results in Figure 4b indicate that our selection policy is indeed shows better performance.

## 5 CONCLUSION

In this paper, we tackle the problem of Active Learning of VLM prompts in a Test Time setting. Building on prior work which demonstrated the relevance of Active Learning in a Test Time setting, we extend that to Prompt Learning in VLMs. We further demonstrate the relevance by showing that marginal entropy minimisation on uncertain samples reinforces errors in the model. This makes knowing the true label even more relevant. Ours is the first work to deal with the problem of Active Test Time Optimisation with a single test sample at a time. We use a dynamically adjusted threshold for entropy based on which we select the samples which will be chosen for active labelling and we do so in such a manner that our annotation budget is not exceeded or exhausted too early into the data streaming process. Our method uses a replacement policy which prioritises class balancing and informativeness of samples to make intelligent use of the limited-size buffer. We propose a fair evaluation protocol which shows that Active Learning is indeed an effective strategy in the Test Time Prompt Learning of VLMs. Future works can extend this by using pseudo labels alongside active labels and effectively combining their information for even better results.

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

# Active Test-Time Prompt Learning in VLMs
## Supplementary Material

## A    BROADER IMPACT

The goal of deep learning is to learn discriminative invariant feature representations. One roadblock in that goal is that models which are trained on a particular dataset often develop a bias towards it via overfitting which is essentially the emergence of spurious correlations. These spurious correlations pose both technical and ethical concerns that have to be mitigated so that deep learning models can be widely used. In the most common scenario, domain generalization helps us in eliminating these spurious correlations by applying certain techniques before the model's deployment. However, it is not always possible to gather data before the model's deployment, and thus, it becomes imperative to instead adapt the model during inference. We do not see any immediate ethical concerns which are raised by this paper as it does not release any specific dataset nor does it use any human as a subject. However, as with any scientific work, it has the potential for misuse, and thus, we support a continued assessment of methods like ours, which advocate test-time adaptation of large models using expert advice.

## B    RELATED WORK

### B.1    ACTIVE LEARNING

Active Learning promotes label efficiency by imposing a label budget. It can be used in a variety of settings to gain knowledge about some aspect that is not already known and the model is uncertain about via an oracle. The queried samples are then sent to an oracle who returns the true label. Other than choosing on the basis of uncertainty (Lewis & Catlett, 1994; Yang & Loog, 2016; Roth & Small, 2006; Holub et al., 2008), in latest works, the model also tries to maintain diversity (Parvaneh et al., 2022; Sener & Savarese, 2017) in its choice of samples to query. There is typically a dynamic buffer that is maintained by the Active Learning algorithm wherein the annotated samples are put. The buffer is enlarged as more samples which are both diverse and informative are added to it. There have been incorporations of Active Learning into Domain Adaptation as well (Prabhu et al., 2021). In (Wang et al., 2022; Kothandaraman et al., 2022) incorporated Active Learning into Source Free Domain Adaptation which, while not directly dependent on the sourced data, are also not suited for continuous data streams, unlike our TTA setting. In (Saran et al., 2023), they take up the task of actively labelling samples in a streaming setting. However, their work significantly differs from ours as they don't continuously adapt their parameters as the data stream progresses. Instead, they reinitialise their parameters to the original parameters after each new data point is acquired. A contemporary work that is more closely related to ours is (Gui et al., 2024), where they provide some foundational theoretical work on Active Test Time Adaptation. However, their work is also significantly different from ours because they assume a batch setting wherein at each timestep, a minibatch comes for inference while we only assume a single sample. They also do multiple gradient updates in each time step while we do only one. Their labelling is also not done in real-time but effectively postponed by placing the unlabelled samples in a buffer from which they have to be selected for active labelling later. This may not always be possible due to privacy and storage concerns where it may not be feasible to postpone the decision of sending a sample to an oracle and instead retaining it. On the other hand, we make the active labelling decision in real-time based on our dynamically adjusted threshold.

### B.2    TEST TIME ADAPTATION

Adaptation of pre-trained models is necessary so that they can be used optimally for the given task at hand. Fully Test Time Adaptation (Nado et al., 2020; Schneider et al., 2020; Sun et al., 2023; Liang et al., 2023) is the highly realistic and practical setting wherein a pre-trained model has to optimise and adapt its parameters to a situation that it faces during inference, that is, amidst real-time use. A prominent example includes that of a self-driving car where the car has to adapt to unforeseen conditions while it is being used. A popular way to achieve fully test time adaptation has been

to update the statistics of the Batch Normalisation layer during inference. In TENT (Wang et al., 2020), the Batch Normalisation parameters are updated using a self-entropy objective. However, TENT makes the batched input assumption. In MEMO (Zhang et al., 2022), they use the more general case of a single test input by taking multiple augmentations of a single image. TPT (Shu et al., 2022) essentially extends the MEMO philosophy to prompt tuning to make VLMs adapt at test-time by updating prompts. Especially relevant to this paper is the newly introduced paradigm of Active Test Time Adaptation Gui et al. (2024), where the model has the option of querying a few samples during test time adaptation. It was found that at a some latency cost as compared to FTTA methods, the ATTA framework provides much superior results, and thus, its use-case may not completely overlap with that of FTTA.

### B.3 Prompt Learning in VLMs

Prompt Learning has been proposed as a method of fine-tuning VLMs in compute-constrained scenarios due to its parameter efficiency. In CoOp (Zhou et al., 2022b) and CoCoOp (Zhou et al., 2022a), the prompts are appended to the textual tokens and help provide context to the input of the VLM instead of finetuning the entire VLM model. Learning good prompts has been shown to dramatically improve the performance of the CLIP (Radford et al., 2021) model. Maple (Khattak et al., 2023)introduced Multimodal Prompt learning where, along with the text encoder, prompts are also learnt for the vision encoder. PromptAlign(Abdul Samadh et al., 2024) extends the multimodal framework of MaPle to a test time setting, and adds Distribution Alignment to this by considering Imagenet to be the proxy source dataset and calculating the alignment loss between different layers of the encoders for each image and precomputed statistics. Prompt Learning has been extended as a transfer learning and adaptation method in various ways and settings. Of special interest to us is the Test Time setting (Shu et al., 2022). The Test Time scenario is highly practical given that foundation models like VLMs are becoming more mainstream and adapting them at test time by optimising a small parameter group like prompts is likely to be the way forward. More recently, in (Bang et al., 2024) they introduce Active Learning to Vision Language models where it is noted that diversity in the form of class-balancing is important for non-trivial gains in VLM performance via Active Learning. However, our method is significantly different from theirs because it is in a test time scenario, whereas theirs was in a supervised learning setting.

## C Potential Applications

For the sake of fair comparison with previous arts, we actively label our samples only after evaluating them. However, it is important to note that, when it comes to practical application, we can also reverse the order. That is, we can ask the expert for their advice and take that as the ground truth. This is especially true because we are not restricted by our methodology to postpone the active labelling decision but instead do so in real-time, unlike (Gui et al., 2024) where they collect the unlabelled samples in a buffer and select which ones to query later. Our single test sample in a time-step assumption makes our setting even more practical.

Ideal scenarios for practical application are high-risk ones like autopilot systems and medical diagnosis, where the extra cost and latency in consulting an expert is justified by the potential avoidance of hazards. In autopilot systems, when the system is uncertain, it can hand over control to the pilot, who then demonstrates the correct way of handling that particular scenario. The system then stores the pilot's actions in memory and uses them to learn the correct way so that it does not need human intervention in a similar scenario in future. In medical diagnosis, whenever the system is uncertain, instead of making a wrong diagnosis, it sends the sample over to a medical practitioner who then provides his expert opinion.

Our average inference latency per sample, when the buffer size is at its maximum, is around 0.63s, which is about 50% more than that of PromptAlign (Abdul Samadh et al., 2024), whose latency we found to be 0.41s when averaged over all 14 datasets. However, these figures must be contextualized in comparison to those from (Gui et al., 2024), where they observed up to almost **10x** increase in latency compared to their baselines.

# D   ALGORITHMS

---

**Algorithm 1** Dynamic Threshold Selection

---

**Require:** $t, N_{queried}, \hat{\mu}_t, \hat{\sigma}_t$

  **if** $t < T_{min}$ **then**

    # Initially select a static threshold till $T_{min}$ number of test samples

    **Output:**$\tau_0$

  **else**

    # $\alpha$ is the query selection percentile

    **if** $\frac{N_{queried}}{t} \geq \alpha$ **then**

      # Select higher value of z ($z_{high}$) if currently over-querying

      $\tau_t = \hat{\mu} + z_{high}\hat{\sigma}$

    **else**

      Select standard value of z with respect to $\alpha$ otherwise $\tau_t = \hat{\mu} + z_{selection}\hat{\sigma}$

    **end if**

    **Output:**$\tau_t$

  **end if**

---

---

**Algorithm 2** Class Balanced Eviction from Buffer

---

**Require:** Unlabeled Dataset $\mathcal{D}_u, Oracle(.)$ $N_{queried} = 0, \hat{\mu}_0 = 0, \hat{\sigma}_0 = 0$

  **for** $t = 1, 2, 3, ..., |\mathcal{D}_u|$ **do**

    $\hat{\mu}_t, \hat{\sigma}_t \leftarrow \hat{\mu}_{t-1}, \hat{\sigma}_{t-1}$

    $\tau_t = \text{Dynamic Threshold Selection}(t, N_{queried}, \hat{\mu}_t, \hat{\sigma}_t)$

    **if** $H(\hat{x}_t) > \tau_t$ **then**

      $y_t = Oracle(x_t)$

      $N_{queried} \leftarrow N_{queried} + 1$

      **if** the buffer is full **then**

        # Select class k, the class with the most samples in $\mathcal{D}_l$

        $m = \underset{k \in \{1,2,...,K\}}{\arg\max} |c_k|$

        **if** $|m| = 1$ **then**

          # Select sample with lowest CE loss in max class

          $j = \underset{i \in \{1,2,...J\}}{\arg\min} L_{CE}(x_i, y_i)|y_i = c_m$

          Remove $(x_j, y_j)$ from $\mathcal{D}_l$

        **else if** $|m| > 1$ **then**

          # Select the max class with least avg CE loss

          $l = \underset{k \in m}{\arg\min} \bar{L}_{CE}(x_i, y_i)|y_i = c_k$

          # Select sample with lowest CE loss in the max class

          $j = \underset{i \in (1,2,...J)}{\arg\min} L_{CE}(x_i, y_i)|y_i = c_l$

          Remove $(x_j, y_j)$ from $\mathcal{D}_l$

        **end if**

      **end if**

      Add $(x_t, y_t)$ to $\mathcal{D}_l$

    **end if**

  **end for**

---

