# OpenReview forum: "ACTIVE TEST TIME PROMPT LEARNING IN VISION- LANGUAGE MODELS"
_ICLR.cc/2025/Conference — ICLR 2025 Conference Withdrawn Submission_

### Official Review · Reviewer_szra · 2024-10-28

**Soundness:** 2
**Presentation:** 2
**Contribution:** 2
**Rating:** 3
**Confidence:** 4

**Summary:**

This paper studies the active test-time prompt tuning for the vision-language model. The authors propose a novel test-time prompt tuning method, outperforming existing test-time prompt tuning methods.

**Strengths:**

1. The integration of active learning with test-time prompt tuning presents a novel approach. However, previous studies have explored the benefits of employing active learning for test-time adaptation.
2. The experimental results are promising, indicating that actively queried labels can effectively enhance test-time prompt tuning.

**Weaknesses:**

1. The novelty of this paper is marginal. The active test-time adaptation, test-time prompt tuning, and data buffer are not novel and have been proposed by existing studies [1-3]. Although some studies have been discussed in this paper. However, the major technical improvement of this paper has not been highlighted.
2. The coefficient of different losses needs a more detailed discussion. Why average performance on four datasets can derive the optimal value of alpha? Do these four datasets consistently achieve the best performance when alpha = 1 and what about other datasets?
3. Do the comparison methods adopt the labeled data from prompt tuning? How do these methods query the ground-truth labels during the testing?
4. The presentation of this paper can be improved. For instance, Table 2 is too large and Figure 3 has some blank space.

[1] Shurui Gui, Xiner Li, Shuiwang Ji: Active Test-Time Adaptation: Theoretical Analyses and An Algorithm. ICLR 2024.

[2] Dingchu Zhang, Zhi Zhou, Yufeng Li: Robust Test-Time Adaptation for Zero-Shot Prompt Tuning. AAAI 2024: 16714-16722

[3]Taesik Gong, Jongheon Jeong, Taewon Kim, Yewon Kim, Jinwoo Shin, Sung-Ju Lee: NOTE: Robust Continual Test-time Adaptation Against Temporal Correlation. NeurIPS 2022

**Questions:**

Please refer to the "Weaknesses" section.

---

### Official Review · Reviewer_AAHJ · 2024-10-30

**Soundness:** 2
**Presentation:** 1
**Contribution:** 2
**Rating:** 3
**Confidence:** 3

**Summary:**

This paper concerns the problem setting, Active Test Time Prompt Learning, which is underexplored yet well-motivated by considerations such as performance, efficiency and real-life applicability. The proposed method successfully combines Active Learning and Prompt Tuning to come up with a test-time optimisation scheme that is evaluated to be an improvement over existing methods.

**Strengths:**

1. This paper first applies Active Learning in a Test Time setting for VLMs.
2. The motivation is well explained.

**Weaknesses:**

1. The paper writing is poor. There are many unclarities in this paper. For example,
(1) For Table 1 you need to provide an explanation for top 2%. Additionally, I suggest including more results for top 1%, 5%, as the improvements presented in Table 1 are marginal and not very convincing.
(2) In Line 087, what are ''non-informative samples'' and ''non-represented classes''? It's difficult to understand these terms without referring to the methods section.
2. The comparison of domain generalization and cross-dataset settings is not very convincing. Why not include the same baseline models in Tables 2 and 3? Furthermore, the improvements achieved by the proposed method appear to be marginal.
3. How does the model perform on standard domain generalization benchmarks, such as PACS and VLCS, which many CLIP-based prompt tuning models [1,2,3] use for evaluating domain generalization?

[1] Bai, S., Zhang, Y., Zhou, W., Luan, Z., & Chen, B. (2024). Soft Prompt Generation for Domain Generalization.
[2] Bose, S., Jha, A., Fini, E., Singha, M., Ricci, E., & Banerjee, B. (2024). Stylip: Multi-scale style-conditioned prompt learning for clip-based domain generalization.
[3] Zhu, L., Wang, X., Zhou, C., & Ye, N. (2023, June). Bayesian cross-modal alignment learning for few-shot out-of-distribution generalization.

**Questions:**

See Weakness.

---

### Official Review · Reviewer_cnhf · 2024-11-03

**Soundness:** 2
**Presentation:** 2
**Contribution:** 1
**Rating:** 1
**Confidence:** 4

**Summary:**

This paper explored the intersection of test-time adaptation (prompt tuning) and active learning for adapting foundation models. The authors propose an uncertainty-based thresholding method to improve the existing test-time prompt tuning (TPT) method based on an empirical observation and advocate the use of a class-balanced buffer-maintaining technique and class-wise representation alignment. The proposed method was validated on few-shot image classification tasks on some specialized domain and domain generalization regimes.

**Strengths:**

The intersection of active learning and test-time adaptation is increasingly crucial, and the authors are timely exploring that field.

**Weaknesses:**

- Weak motivation
  - The authors propose the Entropy-based thresholding method to dynamically determine samples used to update the prompt parameters.
  - The authors claim the necessity of that thresholding technique based on the observation in Table 1.
  - However, the improvement achieved by the thresholding is too minor (different in hundredths place value), which raises a question on the necessity of the proposed method.
- Novelty of the proposed method
  - The authors directly adopt the MaPLe framework [1] as a learning target model, and adopt the PromptAlign [2] loss (which is constructed with the combination of TPT [3] loss with feature alignment loss) with two modifications -- 1) entropy-based filtering, and 2) class-wise alignment with balanced buffer.
  - However, the considered modification is not novel at all as the thresholding technique and the class-wise treatment are just trivial engineering extensions of an existing method. If the author wants to claim novelty here, they should provide at least their unique motivation for leading these components in the active test-time prompt tuning framework whether theoretically or empirically. However, the authors do not provide these (the motivation for the thresholding method is not solid as mentioned above).
- Lack of rigor in experiments
  - Lack of baseline methods: The authors only consider existing prompt tuning methods and incorporation of TPT, and do not provide a comparison with other state-of-the-art active learning methods or test-time adaptation methods.
  - Lack of discussion on computational cost and runtime perspectives: Besides accuracy, discussing the runtime (and computational complexity) of the proposed method and baseline methods is crucial to describe the practical usefulness of the proposed framework.
- Marginal improvement of proposal
  - Overall, the improvements noted in the tables are very marginal (when it comes to the comparison with baseline or ablation studies), which makes the reviewer speculate about the effectiveness of each component in the proposed method and the framework itself.

> Reference
1. MaPLe: Multi-modal Prompt Learning, Khattak et al. 2023
2. Align Your Prompts: Test-Time Prompting with Distribution Alignment for Zero-Shot Generalization, Hassan et al. 2023
3. Test-Time Prompt Tuning for Zero-Shot Generalization in Vision-Language Models, Shu et al. 2022

**Questions:**

See the weaknesses section.

---

### Official Review · Reviewer_cizQ · 2024-11-03

**Soundness:** 2
**Presentation:** 1
**Contribution:** 2
**Rating:** 3
**Confidence:** 4

**Summary:**

This paper addresses the active test-time adaptation problem in visual language models, particularly large-scale pre-trained models. First, the authors argue that model updates should be based on confident samples, discarding less confident ones. Building on this insight, they propose a dynamic threshold selection algorithm that identifies the most valuable samples for adaptation. The authors demonstrate that their algorithm outperforms prior prompt-learning algorithms across 10 datasets on average and further exhibits domain generalization capabilities on the ImageNet dataset.

**Strengths:**

1. Proposes a framework for prompt learning in vision-language models.

2. Shows reasonable performance gains.

**Weaknesses:**

1. While the paper presents performance improvements and novel framework for vision-language models, its novelty is somewhat limited, specifically restricted to the threshold adaptation method.

2. Given the focus on test-time adaptation, it would be beneficial to evaluate performance on novel classes, in my view. However, the authors only report results for target classes. Is the framework applicable to novel class analysis, as explored in other works like CoCoOp?

Comments (Minor)

1. the margin in Table 2 should be adjusted
2. The font size of each figure needs to be enlarged, as they are difficult to read.
3. Does TTA represent "test time adaptation"? If so, this is not clarified in the manuscript. Only "TTT" is mentioned, which could lead to confusion.
4. Using "Ours" in the result tables could be misleading. It might be clearer to state that the implementation is based on PromptAlign.

**Questions:**

I noticed that an active prompt-learning method exists, but it is not compared in this paper. Is there a reason this method was omitted?

---

### Note · Authors · 2024-11-15

I have read and agree with the venue's withdrawal policy on behalf of myself and my co-authors.